# A Preliminary Assessment of Health and Safety in the Automobile Industry in Brunei Darussalam: Workers’ Knowledge and Practice of Organic Solvents

**DOI:** 10.3390/ijerph192315469

**Published:** 2022-11-22

**Authors:** Hazimah Hasylin, Khadizah H. Abdul-Mumin, Pg-Khalifah Pg-Hj-Ismail, Ashish Trivedi, Kyaw Naing Win

**Affiliations:** 1Pengiran Anak Puteri Rashidah Sa’adatul Bolkiah Institute of Health Sciences, Universiti Brunei Darussalam, Gadong BE1410, Brunei; 2School of Nursing and Midwifery, La Trobe University, Bundoora, Melbourne, VIC 3086, Australia; 3Occupational Health Division, Ministry of Health, Bandar Seri Begawan BB2313, Brunei

**Keywords:** occupational, environmental, health, organic solvents, automobile, workshop, spray painting, Brunei Darussalam

## Abstract

Automobile industries worldwide extensively use organic solvents. Yet, limited studies have examined the health and safety of handling these solvents, which can only be assured if workers have appropriate knowledge and demonstrate safe practices. A cross-sectional study was conducted to preliminary explore the knowledge and practice among workers who are involved in handling organic solvents in the automobile industry in the largest urban district in Brunei Darussalam. Qualitative data were sourced from open-ended questions, observations, and pictorial evidence through still photographs. Quantitative analysis showed that 75% of the workers practice reading labels, 94.1% use fully covered clothes, 82.4% wear gloves, and 98.5% practice proper hand washing. Furthermore, 98.5% of workers cover container lids. The qualitative analysis illustrated that workers have general knowledge of materials containing solvents but could not identify the exact solvents, the harmful effects of the solvents, pathophysiology, and harmful effects on specific body systems. Health and safety were found to be practiced, albeit not consistently. Health and Safety Authorities in Brunei Darussalam must review and enforce specific policies on the use of organic solvents so that they can be practiced consistently and safely in the automobile industry. Cooperation and collaboration in adhering to the policies are mandatory to ensure health and safety at work.

## 1. Introduction

Organic solvents are carbon-based solvents, which are substances that dissolve another substance to create a homogenous solution [1]. Common organic solvents include isopropanol, toluene, xylene, and solvent mixtures such as white spirits and chlorinated solvents [2]. Depending on the exposure and concentration, organic solvents have harmful effects on the respiratory system, integumentary system, and digestive system, primarily on the liver and kidney, cardiovascular system, and central nervous systems, such as cognitive and emotional deficits, diseases or even death [3]. It has been reported worldwide that a deficit of health and safety procedure in handling these organic solvents lead to adverse health effects. Reducing exposure levels as well as safe practices for solvent use is important in reducing the risk of ill-health effects. Safe practices depend on having the appropriate knowledge of the health risks associated with the exposure and harmful effects of organic solvents [4]. 

Organic solvents are also extensively used in activities such as degreasing, cleaning, and repairing car engines in the automobile industry [5]. Repairing requires painting and coating using paint, varnish, and rustproofing, all of which contain organic solvents [6]. N-hexane is mainly used in solvents, glues, spray paints, coatings, and silicones which is well known for its neurotoxicity and increased risk of development for peripheral neuropathy. Many cases of N-hexane-related neurotoxicity have been reported in the automotive repair industry in Asia, Europe, and the United States [7]. The global N-hexane market showed a compound annual growth rate of 6.6% from 2021 to 2022. However, a recent publication from the United States among automotive technicians showed limited evidence of association of peripheral neuropathy with solvents, especially N-hexane exposure, where exposure to N-hexane, as well as other solvents, were well below their exposure limits [8,9]. Other agents are toluene and acetone, which are found abundantly in commercial thinners and where toluene is a well-known neurotoxic agent. Acetone amplifies or potentiates the neurotoxic effect of N-hexane or other organic solvents [10,11,12,13].

Road Traffic Accidents (RTA) require major repairs, which are mostly conducted in auto garages and typically include damaged body parts such as bumpers and hoods, hammering out and patching up dents, and other minor or major body damage in the automobile industry. In Brunei Darussalam (henceforth: Brunei), for the year 2016, the recorded number of vehicles was 182,238, with 3375 road traffic accident cases reported, giving a ratio of RTA to vehicles at 1:53 [14]. Despite the wide use of organic solvents in mobile auto industries, there are still limited studies that examine the knowledge and practice of these industries’ workers with respect to organic solvents.

There are several existing and related pieces of legislation under the purview of the Safety Health and Environment National Agency (SHENA) that regulates and oversees the health, safety, and welfare of workers in Brunei. Under the Workplace Safety and Health Order (WSHO) 2009, there are special provisions for hazardous substances, and the WSHO specifies the duty of the employer as well as suppliers and provides information for its safe use at the workplace [15]. This body of legislation is provided to principally eliminate or control workplace hazards and risks, and protect the welfare of employees during employment. Although there is a specific clause that lists chronic benzene poisoning in the WHSO, there are no other specific provisions concerning the handling of organic solvents, nor available data or evidence that determines whether workers are compliant with safe practices whilst handling organic solvents. The objective of this study is to explore the knowledge and practice among workers who are exposed to organic solvents in selected auto garages in Brunei.

## 2. Methods

### 2.1. Study Design, Setting, and Sampling

Ethical clearance for this research was approved by the Pengiran Anak Puteri Rashidah Sa’adatul Bolkiah Institute of Health Sciences Research Ethics Committee (IHSREC) (Ethics Reference Number: UBD/IHS/B3/8). A cross-sectional, mixed-method study was conducted in four of the leading automobile workshop industries in the most populous and most urban districts in Brunei. Although all workers who are exposed to or associated with the handling of organic solvents in the automobile workshops were included in this study, Brunei has a population of less than half a million. Hence, of the total of 91 workers in the four workshops: 13 workers were involved during the questionnaire pre-testing, and 68 workers participated in the main study.

### 2.2. Data Collection and Analysis

#### 2.2.1. Questionnaire

Data was collected using a pre-designed questionnaire derived from the questionnaire used for a study on “Knowledge, Attitude, and Practices” (KAPs) regarding organic solvents among printing workers in Hong Kong [16]. The questionnaire was modified for appropriateness to the local setting and focused on knowledge and practice with permission from the researchers. The research team acted as an expert panel group (three experts in environmental health and two experts in public health) and evaluated the suitability of the questions. Following that, two focus groups (six in one group and seven in another) involving 13 automobile workers were conducted for test-retest that aimed at content validity. The words used in the questions were modified so as to be comprehensible to the respondents until there was a consistent understanding of the questionnaire from one respondent to another. 

The final instrument consisted of close-ended questions about current knowledge and practice of organic solvents in the automobile industry. Open-ended questions were also inserted that added qualitative value to the study. The questions aimed to acquire in-depth elaboration that assessed knowledge of adverse health effects of organic solvent exposure. Two researchers took turns collecting data at each study site. The questionnaire was interviewer-assisted with the help of a professional translator who worked at the study site, as a large number of the workers were foreigners and did not speak English. 

The questionnaire comprised questions of a suitable education level and was classified based on the International Standard Classification of Education (ISCED) [17]. The best five safe practices were defined as reading chemical labels, using fully covered clothes, wearing gloves, hand washing, and covering the lids of containers after use. Participants were categorized into direct and indirect exposure to organic solvents based on their job descriptions. Painters who are directly exposed to organic solvents are those involved with body repair workers, mechanics, and panel beaters. Those workers who are not directly exposed include site supervisors and technicians who do not spend time handling the organic solvents [18]. The quantitative data were analyzed using Statistical Package for the Social Sciences (IBM SPSS Statistics Software version 26.0). Frequencies and percentages were used to present categorical variables. 

Qualitative data were tabulated using a Microsoft EXCEL spreadsheet and analyzed using thematic analysis. The process of coding and theme formation ensued until the finalization of themes was undertaken by constantly comparing data within the same questionnaire and with other questionnaires [19]. Numerous discussions were held, and agreement was sought among the researchers on the final themes. 

#### 2.2.2. Observations (Pictorial Evidence)

Additionally, on-site observations and pictorial evidence from still photographs were utilized to further compare and verify responses to the questionnaire. Different days were set for observations and the taking of still photographs showing the handling and storing of organic solvents in the automobile workshops. Photographs of observations of the auto-garages were interpreted and confirmed by all researchers. The photographs were used to corroborate the results of the questionnaire.

## 3. Results

### 3.1. Quantitative Analysis 

Figure 1 and Table 1 illustrate the quantitative findings of this study.

In Figure 1, most workers (76.5%) cited lungs as the organ affected by exposure to organic solvents, followed by skin (63.2%), heart (36.8%), and liver (32.4%). 

Table 1 illustrated that there were 68 workers who participated in this study which represented a 100% response rate of respondents not involved during the test-retest. The mean age of workers was 41 years (range: 22–62 years old). The highest age group was 41–50 years (35.3%). 55.9% of the workers had primary-level education, and only 8.8% had tertiary education level. Further socio-demographic and work characteristics revealed that 39.7% of the workers were current smokers, while 39.7% consumed alcohol. 35.3% of workers were painters who were directly exposed to organic solvents, 17.6% were body repair workers, 20.6% were mechanics, 7.4% were panel beaters, and 19.1% were site supervisors and technicians who were only indirectly exposed. 

Table 2 represents the analysis of knowledge and practice of Health and Safety in handling organic solvents based on the ethnicity, age group, and education level of participants. All three variables showed a non-significant difference between knowledge and practice. 

Figure 2 shows that the majority of the workers were expatriates (58.8%). Expatriates were from Indonesia (15.4%), the Philippines (38.5%), Thailand (28.2%), and others (17.9%), including India and Sri Lanka, and Bangladesh. 

Figure 3 showed that 75% of the workers mentioned they practice reading chemical labels, 94.1% practice using fully covered clothes, 82.4% practice wearing gloves, 98.5% practice hand hygiene, and 98.5% practice covering lids of the containers while handling the organic solvents. 

### 3.2. Qualitative Analysis

Open-ended questions elaborated on the respondents’ “Knowledge and Practice” towards Health and Safety in the handling of organic solvents before, during, and after spray painting and handling car repairs. Qualitative analyses of the respondent’s answers to the open-ended questions indicated three common themes; Theme 1: Awareness of materials containing organic solvents; Theme 2: Harmful effects of organic solvents; and Theme 3: Safety practice and precautions. The themes formulated from the open-ended questions in the questionnaire and findings from the pictorial evidence are described below.

#### 3.2.1. Theme 1: Materials Containing Organic Solvents

This theme represents the participants’ ability to identify organic solvents that they use at their workplace. The majority had limited knowledge of the exact organic solvents that they dealt with in their workplace but managed to identify materials/chemicals that contain organic solvents. Thinner was the most commonly reported material containing organic solvents that workers used for mixing with spray paints or for cleaning spray paints, or for use after car repair. Degreaser and other paint materials were also identified as containing organic solvents, especially by spray painters. 

Some participants specifically highlighted benzene, greases, synthetic oil, engine oil, brake fluids, and gearbox oils as the chemicals that contain organic solvents used in repairing cars. 

#### 3.2.2. Theme 2: Knowledge of the Harmful Effects of Organic Solvents

This theme illustrated the participants’ knowledge of the harmful effects of organic solvents. Participants were able to report the generalized effects; however, they were unable to identify the specific harmful effects of the organic solvents on health. The majority of participants reported the effects on the skin and lungs, such as skin irritation and lung disease or cancer. There were some participants who identified the effects of organic solvents on the liver and kidney as well as the psychological effects of organic solvents. Participants also mentioned the mode of contact, such as skin contact or inhalation, that may lead to harmful effects. However, they were unable to identify the specific organic solvents that may cause harmful effects on the particular organ, system, or body part.

#### 3.2.3. Theme 3: Safety Practice and Precautions

This theme outlined the participant’s attitude and practice of safety and precautions at the workplace. The majority identified safe practices and precautions. Wearing PPE was the most common safety practice reported by participants, despite being low on the scale of the hierarchy of control measures.

Participants, especially those in a supervisory role, mentioned the importance of reading the material safety data sheet (MSDS) for the proper handling of the materials. Similarly, they also identified safe practices such as the use of local exhaust ventilation as well as carrying out spray painting in a properly designated and isolated area. 

Some of the participants also pointed out the importance of cleaning and hygiene measures after spray painting and repair activities, including dealing with any paint/chemical spillage. However, improper practices of using thinner as a cleaning agent for hands during and after work, as well as removing their gloves for better grip during the work, were also mentioned by some of the participants (Table 3). 

### 3.3. Pictorial Evidence

Observation and pictorial evidence (Figure 4) identified that some of the chemicals used in the workshops had product labels with detailed information about their contents, including the organic solvents as well as their harmful effects. However, participants were only able to mention the name of the material used and the general health hazards. They were not aware of the organic solvent in that material and, similarly were unaware of the specific health hazards caused by them (Figure 4a).

Based on the pictorial evidence (Figure 4a): the following chemicals and organic solvents were used; Jotun paints, G6 paints, Kimton, Sikkens Autobase Plus, Acrylic Thinner, Acetone, Methyl Ethyl Ketone (MEK), and Spirits.

Pictorial evidence in the workshops also identified a designated/isolated area for spray painting activity with an exhaust ventilation system installed (Figure 4b). However, in two of the workshops, pictorial evidence showed inconsistencies in the appropriate or accurate usage as well as a choice of PPE; i.e., spray painters were not using fully covered clothes as well as inappropriate use of the mask/respirator, i.e., wearing of a surgical face mask instead of using a proper chemical respirator (Figure 4c). 

Figure 4d indicated an improper procedure for dealing with paint spillage on the floor, i.e., spilled paint was being washed into the drain led to the public drainage system outside the workshop building, as well as improper storage of chemicals after they had been used. 

## 4. Discussion

Lack of health and safety controls and measures while working with organic solvents can adversely affect multiple body organs and systems. This study was conducted to assess awareness as well as work practices among automobile workers while handling organic solvents at the workplace. 98.5% of participating workers were male, showing that males dominate automobile workshops in Brunei Darussalam, which is similar to automobile industries in other countries such as Australia, Japan, and Europe [20,21,22]. More than half of the workers in this study were of primary level education (55.9%) and comprised expatriates (58.8%) from Indonesia, Philippines, Thailand, India, Sri Lanka, and Bangladesh. It is not unusual to have diverse expatriate groups from many parts of the world. This can be attributed to globalization, whereby people migrate to another country due to overpopulation of the workplace in their home countries or where there are no opportunities available in the workplace. In 2017, the number of international migrants was estimated to be 258 million. According to the ILO, about 63% (164 million) of international migrants are migrant workers, of whom 58.4% are men. 61% of all international migrant workers reside in North America (23%), northern, southern, and western Europe (24%), and the Arab States (14%) [23]. The diversity in nationalities and background experience may also have a significant impact on the respective workers’ understanding, knowledge and safety practices with respect to the use of organic solvents.

The quantitative findings indicated that automobile workers’ knowledge about the harmful effects of organic solvents was superficial and general, where 52 (76.5%) and 43 (63.2%) participants, respectively, were able to mention the lung and skin as the organs affected by exposure to organic solvents. Participants were unable to identify the exact organic solvents that can cause specific harmful effects to different body organs and systems. However, with respect to following safe practices, more than 94% of participants wear fully covered clothes, practice hand hygiene and cover the lids of containers, while 82.4% wear gloves and 75% read chemical labels, as illustrated in Figure 2 when handling organic solvents at their workplace. The PPE practice, hand hygiene, and reading the safety data sheets (SDS) at work were also mentioned by participants in the qualitative part of the study. Participants, especially in a supervisory role, also mentioned the importance of reading the material safety data sheet (MSDS) for the proper handling of materials.

However, improper safety practices, such as using thinner for cleaning hands, as well as removing gloves for better grip during work, were also mentioned. Moreover, the site visit and pictorial evidence were not congruent with the participants’ responses and were evidence of the improper use and choice of PPE, i.e., wearing a surgical mask and not using full protection body coveralls whilst spray painting. As per ILO recommendations, depending on the job task and exposure, workers should be provided with proper protective overalls, footwear, respiratory protection that filters out chemical vapors, and equipment to protect the face, eyes, and hands whilst handling chemicals, including spraying operations [24]. As mentioned by participants in this study, the use of solvent for washing paints was also observed among painters in a study conducted in Ghana [4]. As part of health and safety precautions, it is important to ensure that the right equipment and tools are used and are in good condition. The effective control of chemical exposure during spray painting includes the proper selection of spray-painting equipment, a properly designed and ventilated spray-painting booth, and the use of the correct PPE [25]. In Brunei, safety practices and procedures relating to work are provided under the Workplace Safety and Health Order [15]. In this legislation, it is stated that the general duties of the employer include the provision of PPE, providing equipment for the convenience of employees, ensuring that all machinery, equipment, and working environments are safe at all times, providing adequate information and procedures that are necessary before carrying out any work as well as being responsible for the repair and maintenance of working environments that may pose a risk and result in accidents and injuries at the workplace. Based on our observations, basic training included donning and doffing of PPE, as well as basic use of respirators and gloves.

This research is unique in that addition to the quantitative descriptive cross-sectional research. It also employed open-ended questions and pictorial evidence. The open-ended questions showed that auto-garage workers had good knowledge of the materials/chemicals being used at their workplace that contained organic solvents. However, the majority had limited awareness about the actual solvents they were dealing with at the workplace. Similarly, participants were able to mention the generalized harmful effects of organic solvents, such as damage to lungs or skin following inhalation or skin exposure, with some respondents also mentioning the harmful effects on the kidney and liver as well as on mental health. However, they had a poor awareness of how specific solvents affect particular body organs or systems.

By contrast, from the site observation and the pictorial evidence, it was demonstrated that product labels on the material/chemical containers/bottles provided detailed information about its ingredients, including the organic solvents used as well as their harmful effects. There is a discrepancy in the quantitative and qualitative analysis, as shown by the pictorial evidence. The analysis indicated that although a worker may have the knowledge and state that they employed good safety practices, what they said may not necessarily be the case in reality [23,26]. Qualitative analysis of the open-ended questions further indicated the importance of cleaning the workplace area, disposal of used paints, and hand washing practice after their work activities. Hand washing after work, especially before eating or drinking is considered an important measure for preventing chemical exposure at work [24]. 

In this study, the sample size was small. However, this was generally acceptable for a Brunei setting, as it has a low population. The language barrier presented a challenge, as well as the literacy levels of participants as a barrier for data collection, but these were addressed by the presence of a professional translator at the respective study sites for translation and transliteration. The use of a mixed-methods study through cross-sectional design, responses from open-ended questions, and pictorial evidence through site observations ensured data triangulation that complemented and strengthened the quantitative findings. 

## 5. Conclusions

This study is the first of its kind to explore knowledge and practice of health and safety while handling organic solvents amongst automobile workshop workers in Brunei. The results provided baseline information regarding prevailing awareness of the harmful effects of solvents, as well as the safety practices of automobile workers, for improving health and safety at work. This information can be used to develop policies that benchmark the handling of organic solvents at automobile workshops. Training sessions and seminars should be held regularly for the workers to refresh their knowledge and practice and raise awareness of the health and safety measures in handling organic solvents. Provision of the necessary information is deemed imperative, e.g., providing the Material Safety Data Sheet (MSDS) for the relevant chemicals, which in turn can help to improve safety practices among workers. Health and Safety authorities in Brunei Darussalam should regularly review and enforce specific policies for organic solvents so that they can be practiced consistently in the auto-garage industry. When coupled with existing legislation, a designated national agency could be set up so that cooperation and collaboration between government and workplaces can enhance the health and safety measures at such workplaces.

## Figures and Tables

**Figure 1 ijerph-19-15469-f001:**
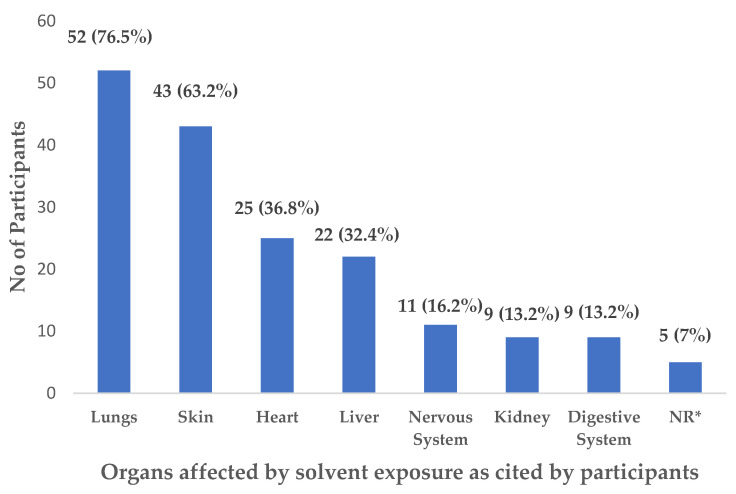
Organs affected by solvent exposure. (Answer more than one). NR* Five workers did not respond to this question.

**Figure 2 ijerph-19-15469-f002:**
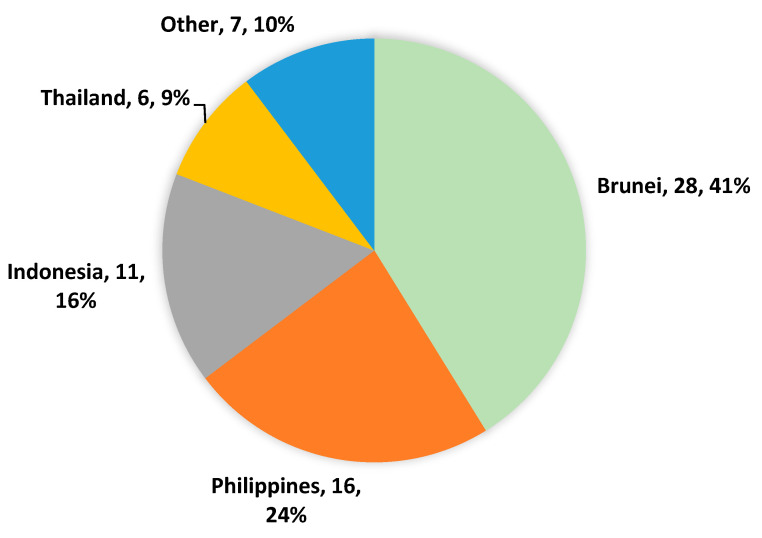
Nationalities of the Participants.

**Figure 3 ijerph-19-15469-f003:**
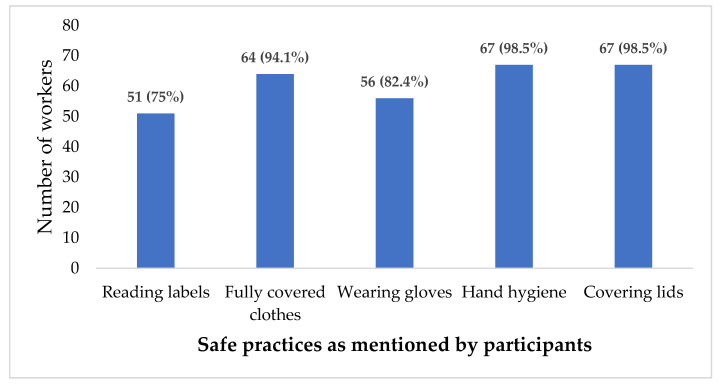
Five best safety practices of automobile workers at their workplaces in Brunei Darussalam (Workers who replied “Yes” to the question on following these practices).

**Figure 4 ijerph-19-15469-f004:**
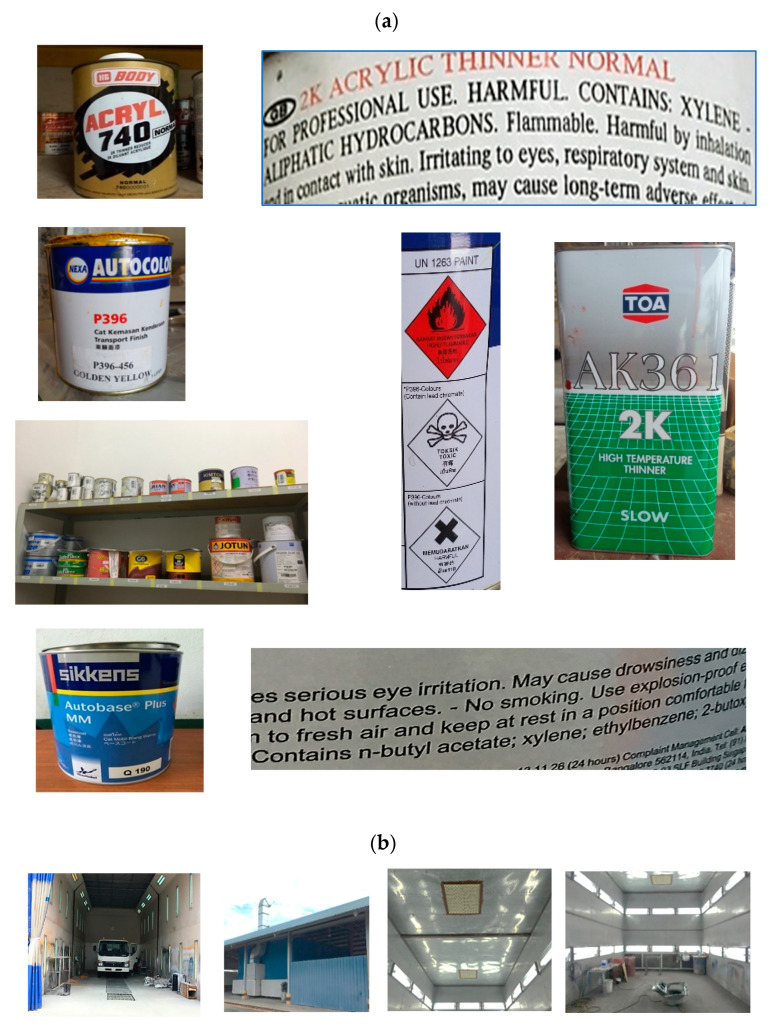
(**a**) Product details with ingredients (organic solvents) and harmful effects; (**b**) Designated spray paint area with a Local Exhaust Ventilation (LEV) System; (**c**) Improper PPE practices at the workplaces; (**d**) Improper storage and poor spillage management.

**Table 1 ijerph-19-15469-t001:** Socio-demographic and Work Characteristics of Study Participants.

Characteristics	Number (%)
Age Groups (years)	
<30	8 (11.8%)
30–40	21 (30.8%)
41–50	24 (35.3%)
≥50	15(22.1%)
Gender	
Male	67 (98.5%)
Female	1 (1.5%)
Ethnicity	
Malay	9 (13.2%)
Chinese	19 (27.9%)
Others	40 (58.8%)
Education	
Primary	38 (55.9%)
Secondary	24 (35.3%)
Tertiary	6 (8.8%)
Smoking	
Current Smoker	27 (39.7%)
Ex-Smoker	09 (13.2%)
Non-Smoker	32 (47.1%)
Alcohol	
Consumes alcohol	27 (39.7%)
Past alcohol consumption	09 (13.2%)
No alcohol	32 (47.1%)
Work Activity	
Painter	24 (35.3%)
Body repair	12 (17.6%)
Mechanic	14 (20.6%)
Panel beater	5 (7.4%)
Others	13 (19.1%)

**Table 2 ijerph-19-15469-t002:** Comparison of Knowledge and Practice of Health and Safety in Handling Organic. Solvents with Ethnicity, Age, and Education level of participants.

		Frequency (%)			*p*-Value
		Yes	No	Total	
Ethnicity	Malay	9 (14.3%)	0 (0%)	9 (13.2%)	0.243 ^b^
	Chinese	19 (30.2%)	0 (0%)	19 (30.2%)	
	Others	35 (55.6%)	5 (100%)	40 (58.8%)	
Age group	20–24	3 (4.8%)	0 (0%)	3 (4.4%)	0.742 ^b^
	25–29	5 (7.9%)	0 (0%)	5 (7.4%)	
	30–34	11 (17.5%)	1 (20%)	13 (19.1%)	
	35–39	7 (11.1%)	1 (20%)	8 (11.8%)	
	40–44	9 (14.3%)	1 (20%)	10 (14.7%)	
	45–49	13 (20.6%)	1 (20%)	14 (20.6%)	
	≥50	15 (23.8%)	0 (0%)	15 (22.1%)	
Education level	Low education	37 (58.7%)	1 (20%)	38 (55.9%)	0.107 ^b^
	Medium education	20 (31.7%)	4 (80%)	24 (35.3%)	
	High education	6 (9.5%)	0 (0%)	6 (8.8%)	

^b^ fisher exact test.

**Table 3 ijerph-19-15469-t003:** Knowledge of organic solvents, Knowledge of harmful effects of organic solvents, Safety practices, and precautions whilst handling organic solvents (Qualitative answers).

Factors	Questionnaire	Participants	
		N	(%)
Knowledge of organic solvents	“Most of the organic solvents which we are using now are the ingredients in the Lacquer thinner that we mix with the spray paints.”	52	76.47
	“I use thinner to mix with car paints that contain organic solvents; it is for clear, hardener and appears to get more glossy car paints.”	5	7.35
	“The organic solvents are in the materials that I deal with, such as thinner, degreaser, colour paint materials, and other volatile organic compounds such as engine oil and greases.”	56	82.35
	“My work is repairing engines. Some solvents that I have used during work include; greases, synthetic oil, engine oil, brake fluids and gearbox oil. I also use benzene to clean engine parts to make them shine.”	2	2.94
Knowledge of harmful effects of organic solvents	“It causes skin irritation if spilled or if in contact with organic solvents. Lungs defects can happen if we inhale.”	53	77.94
	“Organic solvents can damage the kidneys, lungs and skin. Skin can be very itchy and dry if in contact. Small particles from solvents may be inhaled that can either damage the lungs or cause lung cancer if there is long-term exposure.”	9	13.23
	“The solvents may possibly be inhaled through the nose going to the lungs. Hand chemicals such as thinner also can affect the skin causing skin disease. The long-term use of organic solvents may affect the liver.”	55	80.88
	“Organic solvents cause skin irritation through direct contact, leading to lung problems due to inhalation”.	5	7.35
	“Organic Solvents can cause hallucinations if exposed to over the long-term and suffocation if inhaled in the case of working without proper respiratory protection.”	21	30.88
Safety Practices and Precautions	“Safety is the first thing before starting painting work. It is better to know the safety procedures; like wearing PPE such as safety goggles, boots, gloves and mask (half-face for preparation stage and full-face during painting).”	2	2.94
	“We use disposable masks for daily car repairs so that we will prevent ourselves from inhaling dangerous chemicals like solvents used in car repairs. We also use a full-face respirator (half-face with gas canister) to minimize the paint exposure. We check the gas canister 3 times a week.”	50	73.52
	“Every time we work in the automotive industry and use chemical materials, we need to follow the instructions for each item.”	40	58.82
	“The Local Exhaust Ventilation (LEV) must be switched on during spray painting to prevent inhalation of dust from spray paints. We also need to wear rubber gloves, a respirator and safety shoes whilst painting.”	6	8.82
	“Make sure that surrounding places are clear and working must not be in open places. The painting room must be well ventilated with equal air in and air out. Wear safety goggles and a mask. All doors must be locked and the LEV switched on. Prepare air breathing systems and painting booth ventilation equipment, organize the requisition of paint and solvents.”	39	57.35
	“The ‘No Entry’ sign must be switched on while performing spray painting or heating cars using an oven. This is to prevent other people coming in during the procedure and to prevent inhaling dust from the spray paint that contains harmful organic solvents.”	11	16.17
	“Prepare sand to clear up any oil spills. Used oil to be absorbed with chemicals. Paint sticking on paint brushes are cleaned up using lacquer.”	39	57.35
	“Wash hands with soap because if we use thinner, then it can cause irritation to the skin. Used oil after car repairs must be properly disposed of.”	57	83.82
	“PPE is cleaned. Dispose of one time used items. Clean the workshop area. Wash hands after finish working. Tidy things up, and keep all tools and chemicals in a cabinet. For the painting room; it needs to be cleaned and unwanted chemicals also need to be cleared and disposed of. We segregate the chemicals, plastic, paper and used oil properly and they are disposed of properly.”	35	51.47
	“We use special soap provided by the company to wash our hands. Occasionally, we also use thinner to clean off paints and grease from the hands.”	48	70.58
	“If it is difficult to perform my work, I will remove my gloves.”	2	2.94

## Data Availability

The datasets generated and/or analyzed during the current study are not publicly available due to restrictions on intellectual property regulations of the Research Ethics Committee.

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
