# Peer review of "A Preliminary Assessment of Health and Safety in the Automobile Industry in Brunei Darussalam: Workers’ Knowledge and Practice of Organic Solvents"

_ijerph, 2022, doi:10.3390/ijerph192315469_

Round 1

Reviewer 1 Report

This paper deals with knowledge and practices of workers in automobile industry regarding the use of solvents in Brunei Darussalam.

The paper is well-written overall. The methods and findings of the paper are clearly presented and the results are interesting.

However, the results are only descriptive. An analytical analysis should identify possible differences according to education level, nationality, age .....

Reference number [7] was published in 2001. It would be relevant to check whether N-Hexane is still widely used in the US and European countries. Is it still used in Brunei?

The authors mentioned that « There were also participants who identified the effects of organic solvents on the liver, kidney as well as the psychological effects. However, they were not able to identify the specific organic solvents that may cause the harmful effects on the particular organ, system or body part. ». Actually, the level of knowledge of these workers is quite high. Do you think that all physicians are aware of the specific health effects of solvents?

Please provide the questionnaire used to collect the data.

Figure n°1: I don't understand the graphical representation in a pie chart when each participant could have given several answers. The sum must be more than 100%. Please give the exact numbers.

The following sentence needs to be reworded « There is study a small sample size in this study, but this is generally acceptable for Brunei setting, being a mal country. »

This sentence sounds strange “It is not unusual to have diverse expatriates’ workers in many different parts of the world » Did you mean « from many parts of the world.. »

Author Response

Thanks very much for your comments. Kindly find attached the point-by-point response to your suggestions.

Reviewer 2 Report

The manuscript addresses an important topic but as stated in the narrative, the sample size is very small.  The authors believe that this is acceptable due to the small population of workers within the country.  Regardless, the data do not allow for any real quantitative interpretation.  Since the questionnaire was not provided, it was impossible to determine the quality of information gathered and its usefulness.  The information did not provide the reader any insight to likelihood of actual issues with exposures based on the specific hazards of a given exposure. If accepted, the title should be changed to indicate that the manuscript is a preliminary assessment, and the narrative should include this information.

It is suggested that the manuscript be extensively edited for clarity and presentation and resubmitted.

Author Response

Thanks very much for your comments. Kindly find attached the amendments as per your comments.

Reviewer 3 Report

The paper is a simple study carried out to evaluate the possibile lack of knowledge in workers of automobile industry in Brunei.

The text need some changed to be more clear and complete.

First: the references must be more specific within the text (for example: in the second page, line 50-52 you talk about the neurotoxicity of toluene and acetone, the reference is a paper on oxidative stress. It’s true that in the same paper authors talk about neurotoxicity of toluene but using another bibliographic reference, furthermore a neurotoxicity for acetone, for me is quite unknown, you need to use a specific reference for this sentence.

Be careful in the text,  there are numerous typos.

Showing the fig. 1 it is useful to explained what are the correct answers.

In Results: please do not repeat the same data you showed in tables or figures.

Figure 2 is not clear, the percentage is about workers that said YES to the questions? Please write it

You need to explaine in the text the exact solvents that are used in the workplaces, in particular you need to indicate for each type of workers what are the solvents they are in contact with, and in the light of this information you can evaluate if, for example, the data in figure 1 are correct or not.

If available, could be interesting to know what are the environmental concentration levels for the different solvents.

It is necessary to inform the reader if the workers have undergone specific training organised by the employer. This information could lead to really different situations.

Pagg 6-7-8: the answers to questionnaire must be shown using a table or figure, not in a narrative form, and, were appropriate, you need to indicate the correct answer.

I think that for the questionnaire answers you need to consider the different type of workers, and evaluate if a specific task has different output.

Figure 3.5: it is needed less blurry figures.

Line 316: you said that some workers read the safety data sheet: this is a data you need to write with greater details, it is not secondary, this is the main source of information about safety and health risk owing to chemicals. Could be usefull write how many workers read the SDS, and who are they? In which task?

Finally the references are written not in the form requested by the journal, you need to change it.

Author Response

Thanks very much for your comments. Kindly find the amendments made based on your suggestions.

Round 2

Reviewer 1 Report

No additional comments

Reviewer 2 Report

The revisions to the manuscript are acceptable for publicaiton.

Reviewer 3 Report

Currently the text is more clear and the goal of the paper is quite better shown